# The effectiveness of scoring balloon angioplasty in the treatment of chronic thromboembolic pulmonary hypertension

**Masao Takigami, Hideo Tsubata, Naohiko Nakanishi**  ***, Yuki Matsubara, Noriyuki Wakana, Kenji Yanishi, Kan Zen, Takeshi Nakamura, Satoaki Matoba**

Department of Cardiovascular Medicine, Graduate School of Medical Science, Kyoto Prefectural University of Medicine, Kyoto City, Kyoto Prefecture, Japan

* naka-nao@koto.kpu-m.ac.jp

## Abstract

### Background

Balloon pulmonary angioplasty (BPA) is an effective treatment for inoperable chronic thromboembolic pulmonary hypertension (CTEPH). The purpose of this study is to evaluate the therapeutic effect and safety of the non-slip element percutaneous transluminal angioplasty (NSE PTA) scoring balloons in BPA.

### Methods

108 pulmonary artery branches in 14 CTEPH patients who underwent BPA using NSE PTA scoring balloon (the NSE PTA group) or plain balloon (the POBA group) and pressure gradient evaluation were analyzed. We compared the improvement of the pressure ratios after BPA (Δ Pressure ratio) of both groups.

### Results

There was no significant difference in the Δ Pressure ratios of the two groups (0.241 ± 0.196 POBA, 0.259 ± 0.177 NSE PTA, p = 0.63). No complications occurred in the NSE PTA group, while 3 episodes of hemoptysis were seen in the POBA group. This, however, was not found to be significant (p = 0.27). In the cases where balloon-to-vessel ratio exceeded 1.0 (n = 35), multivariate analysis showed that the use of NSE PTA scoring balloon and pressure ratio before BPA were significantly correlated with Δ Pressure ratio (β coefficient: 0.047, 95% CI: 0.0016 to 0.093, p = 0.043 and β coefficient: −0.60, 95% CI: −0.78 to −0.42, p < 0.01, respectively).

### Conclusions

Although NSE PTA scoring balloon was safe, there was no significant pressure gradient improvement with NSE PTA scoring balloon compared to conventional BPA. Nevertheless, the NSE PTA scoring balloon showed effective blood-flow improvement in the case of large balloon-to-vessel ratio.

**Data Availability Statement:** All relevant data are within the paper.

**Funding:** The authors received no specific funding for this work.

**Competing interests:** The authors have declared that no competing interests exist.

## Introduction

Chronic thromboembolic pulmonary hypertension (CTEPH) is a progressive disease associated with increased pulmonary vascular resistance and pulmonary hypertension (PH) as a result of organized thromboses, clinically classified as Group 4 PH [1]. In spite of therapeutic developments, the prognosis of CTEPH remains poor. Pulmonary endarterectomy (PEA) reduces pulmonary arterial pressure and improves symptoms and prognosis in patients with surgically accessible CTEPH [2–5]. For inoperable patients or patients with residual pulmonary hypertension after PEA, pulmonary vasodilators such as soluble guanylate cyclase stimulators are indicated [6, 7]. In recent years, the efficacy of balloon pulmonary angioplasty (BPA) has been reported for the treatment of inoperable CTEPH. BPA has been shown to improve symptoms, exercise tolerance, right heart function, and long-term prognosis in patients with CTEPH [8–10].

In the pulmonary artery of CTEPH patients a mesh or slit-like organized thrombus, known as a web lesion, can form and inhibit pulmonary blood flow [11]. In BPA, a balloon expands and shifts the organized thrombus to the vascular wall in order to improve blood flow [12]. Unfortunately, over-dilatation injures of the pulmonary artery can occur and lead to parenchymal hemorrhage and hemoptysis. Currently, auxiliary devices such as intravascular ultrasound (IVUS) and pressure monitoring devices can be used to reduce such complications [13–15]. At this time, the best way to maximize the therapeutic effect while limiting the complications of BPA has not yet been well established.

Currently, there are many studies showing the usefulness of the scoring balloon in the treatment of coronary artery disease [16, 17]. Non-slip element percutaneous transluminal angioplasty balloon (NSE PTA) is a scoring balloon with three nylon elements and can prevent the balloon from slipping during expansion (Fig 1). It has also been reported that hardened lesions, such as those that are calcified, can be satisfactorily expanded by applying concentrated force to the elements [18]. Additionally, in peripheral vascular treatment, the NSE PTA scoring balloon has been reported to have better dilation and less vessel dissociation [19, 20]. In BPA treatment, NSE PTA scoring balloons may be able to successfully dilate an organized thrombus and obtain improved blood flow, but there have not been any studies on its efficacy and safety thus far.

Therefore, the purpose of this study is to evaluate the therapeutic effect and safety of NSE PTA scoring balloon in CTEPH patients who underwent BPA. We found that although the NSE PTA scoring balloon was safe, there was no significant pressure gradient improvement with the NSE PTA scoring balloon versus the conventional balloon during BPA in most cases. However, the NSE PTA scoring balloon did show superior blood-flow improvement in the cases of large balloon-to-vessel ratios.

## Methods

### Study population and collection of clinical data

This study is a retrospective and single center study approved by The Institutional Review Board of the Kyoto Prefectural University of Medicine. From December 2017 to January 2020,

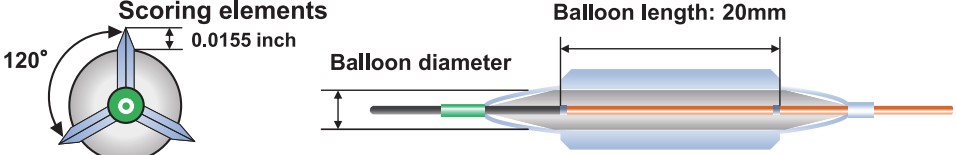

**Fig 1. Non-slip element percutaneous transluminal angioplasty balloon (NSE PTA).** The NSE PTA scoring balloon is a non-slip peripheral angioplasty catheter with three nylon scoring elements for a controlled scoring of the vessel wall and aimed to reduce slipping during balloon inflation.

108 pulmonary artery branches in 14 consecutive patients with CTEPH who underwent BPA using NSE PTA scoring balloon or plain balloon and had pressure gradient evaluation with pressure microcatheter were analyzed. Clinical data, such as age, sex, World Health Organization functional class (WHO-FC), six-minute walking distance (6MWD), brain natriuretic peptide (BNP) and medications, were collected from the electronic medical record at the time of treatment. Hemodynamic characteristics were assessed with right heart catheterization before BPA and right atrium pressure (RAP), pulmonary artery wedge pressure (PAWP), pulmonary artery pressure (PAP) and cardiac output (CO) with thermodilution were measured. Since this study is retrospective observational study, IRB of our institute waived the requirement for informed consent. All data were anonymized for collection.

## BPA procedures

BPA procedures were performed via femoral vein or jugular vein approach. 8-Fr sheath was inserted into the vein and 6-Fr ParentPlus guiding sheath (MEDIKIT, Tokyo, Japan) was advanced to the main pulmonary artery through the 8-Fr sheath using a 0.035-inch wire (Radifocus Guide Wire M; Terumo, Tokyo, Japan). We selected a branch of the pulmonary artery by a 6-Fr guiding catheter (Profit MP, JR4.0 or AL1.0; NIPRO, Osaka, Japan). Pulmonary angiography was performed manually using half contrast medium diluted with saline. A 0.014-inch guidewire (B-pahm; Japan Lifeline, Tokyo, Japan) was crossed under the pulmonary angiography, and IVUS (Eagle Eye Platinum; Volcano, San Diego, CA) was used for the detection of vessel diameter. Vessel diameter was defined as short diameter assessed by IVUS. The balloon size, the expansion pressure and the choice of NSE PTA scoring balloon or plain old balloon angioplasty (POBA) were at the discretion of the operator. Balloon-to-vessel (B/V) ratio was defined as the balloon diameter divided by vessel diameter assessed with IVUS.

## Evaluation of pressure gradient of the target lesion

Distal and proximal pressure readings of the target lesion were measured before and after balloon dilatation using microcatheter with an optical pressure sensor (Navvus MicroCatheter; ACIST Medical Systems, Inc., Eden Prairie, MN). This microcatheter has been reported to show better correlation between microcatheter and pressure wire fractional flow reserve measurements [21]. This information was used to calculate the pressure ratio between the two readings. Since this pressure microcatheter was a rapid exchange type catheter, there was no need to change the guidewire and the pressure gradient could be measured while maintaining the guidewire's position throughout the BPA procedure. The pressure microcatheter on the guidewire was advanced to the tip of guiding catheter where the 2 pressures, a tip of guiding catheter and pressure microcatheter, were equalized after saline flush. Then the pressure microcatheter was advanced and positioned distal to the target lesion. The pressure ratio was defined as the distal pressure of the pressure microcatheter (Pd) divided by the proximal pressure of guiding catheter (Pa) which was recorded before and after BPA (Fig 2). Δ Pressure ratio was defined as the difference between the pressure ratio before and after BPA. Finally, the ratio of pressure ratios was defined as the ratio between the before BPA pressure ratio and the after BPA pressure ratio.

## Statistical analysis

We evaluated whether each valuable was normally distributed. Continuous data were expressed as the mean ± standard deviation (SD). Data that were not normally distributed were presented by median (interquartile range). We compared the baseline characteristics, pressure ratio and Δ Pressure ratio at the target lesion between the NSE PTA group and the

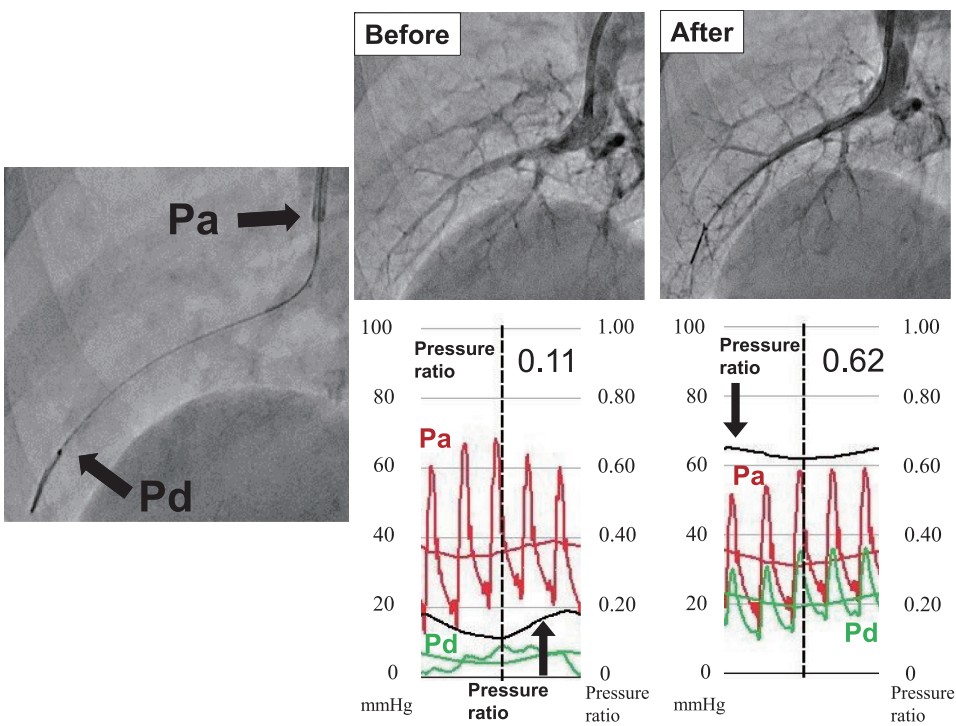

**Fig 2. Assessment of pressure ratio by pressure catheter.** (A) Measuring of blood pressure. Pa was measured by guiding catheter and Pd was measured by pressure microcatheter. (B) Angiography and pressure ratio before and after balloon dilatation. Pressure ratio was defined as Pd / Pa.

POBA group before and after BPA procedure using mixed models for repeated measures. Categorical variables were compared using the chi-square test. Valuables that were not normally distributed were analyzed using non-parametric test. Correlations between pressure ratio before and after BPA in the POBA group and the NSE group were evaluated respectively. We performed univariate analysis using regression analysis with mixed models for repeated measures to find the relationship of Δ Pressure ratio in whole patients and patients with B/V ratio over 1.0 according to Soga's report [22]. Multivariable analysis, adjusting for balloon type, B/V ratio and the pressure ratio before BPA, was performed using multiple linear regression. P-value of < 0.05 was considered statistically significant. The statistical analyses were performed using JMP software version 11.0.0 (SAS Institute Inc., Cary, NC, USA).

# Results

## Patients and procedure characteristics

Table 1 shows the baseline population characteristics of the 14 patients. Mean age was 67.9 ± 15.9 years-old and most of patients were female (93%). Six-minute walking distance (6MWD) was 298.7 ± 100 m. BNP level was 225.3 pg/mL (73.1–495.6 pg/mL) and most of patients had WHO-FC II or III. Eleven patients (79%) were treated by soluble guanylate cyclase stimulator. Hemodynamics parameters showed that the mean pulmonary artery pressure (mPAP) was 39.1 ± 10.3 mmHg, the cardiac output (CO) was 3.6 ± 1.0 L/min and the resulting pulmonary vascular resistance (PVR) was 9.0 ± 4.5 Wood Unit. Fluid balance was well controlled; mean right atrium pressure (RAP) was 8.0 ± 4.0 mmHg, and pulmonary artery wedge pressure (PAWP) was 9.4 ± 3.5 mmHg.

**Table 1. Baseline population characteristics.**

|  | All patients (n = 14) |
|---|---|
| Age, years | 67.9 ± 15.9 |
| Female, n (%) | 13 (93) |
| WHO-FC I/II/III/IV, n | 0/5/7/2 |
| 6MWD, m | 298.7 ± 100.0 |
| BNP, pg/mL | 225.3 (73.1–495.6) |
| Medication, n (%) |  |
| sGC stimulator | 11 (79) |
| Diuretics | 8 (57) |
| Hemodynamics |  |
| RAP, mmHg | 8.0 ± 4.0 |
| PAWP, mmHg | 9.4 ± 3.5 |
| sPAP, mmHg | 67.6 ± 18.2 |
| dPAP, mmHg | 24.4 ± 7.5 |
| mPAP, mmHg | 39.1 ± 10.3 |
| CO, L/min | 3.6 ± 1.0 |
| CI, L/min/m$^2$ | 2.5 ± 0.7 |
| PVR, WU | 9.0 ± 4.5 |

Data are presented as n (%), mean ± SD or median (interquartile ranges). WHO-FC, World Health Organization functional class; 6MWD, 6-minute walk distance; BNP, brain natriuretic peptide; sGC, soluble guanylate cyclase; RAP, right atrial pressure; PAWP, pulmonary artery wedge pressure; sPAP, systolic pulmonary artery pressure; dPAP, diastolic pulmonary artery pressure; mPAP, mean pulmonary artery pressure; CO, cardiac output; CI, cardiac index; PVR, pulmonary vascular resistance.

Procedure characteristics involving the 108 lesions are presented in Table 2. The plain balloon was used in 65 cases (POBA group) and the NSE PTA scoring balloon was used in 43 cases (NSE PTA group). Treated branches and lesion types were well matched between both groups. In the majority of cases, we selected the right lower lobe due to easy access. Most lesions were web lesions (74% in the POBA group, 63% in the NSE PTA group, respectively). Although vessel diameter and balloon size tended to be smaller in the NSE PTA group, there did not appear to be statistically significant differences. Balloon pressure was higher in the NSE PTA group (8.0 (6.0–8.0) atm in the POBA group, and 8.0 (8.0–8.0) atm in the NSE PTA group, p = 0.01). Overall, balloon to vessel (B/V) ratio was similar between two groups (0.93 ± 0.13 in the POBA group and 0.93 ± 0.15 in the NSE PTA group, p = 0.92).

## Pressure ratio of target lesion before and after BPA

Fig 3 shows the relationship between pressure ratios before and after BPA in the POBA group and the NSE PTA group. Both groups had a positive correlation between the pressure ratio before BPA and the pressure ratio after BPA (r = 0.68 in the POBA group; p < 0.01, r = 0.70 in the NSE group; p < 0.01). In addition, a negative correlation was shown between the pressure ratio before BPA and the degree of improvement in pressure gradient ratio (Δ Pressure ratio) (r = −0.80 in the POBA group; P < 0.01, r = −0.54 in the NSE group; p < 0.01); however, there was no significant differences between two groups (p = 0.26). These results suggest that severe lesions have a larger pressure gain after BPA procedure.

Table 3 shows a comparison of distal to proximal pressure ratios before and after BPA. Although pressure ratio before BPA tended to be lower in the NSE PTA group, there was not

**Table 2. Procedure characteristics.**

| | POBA (n = 65) | NSE PTA (n = 43) | P value |
|---|---|---|---|
| Treated site | | | 0.56 |
| Right PA | | | |
| Upper lobe, n (%) | 8 (12) | 10 (23) | |
| Middle lobe, n (%) | 10 (15) | 10 (23) | |
| Lower lobe, n (%) | 28 (43) | 13 (30) | |
| Left PA | | | |
| Upper lobe, n (%) | 2 (4) | 1 (2) | |
| Lingular, n (%) | 5 (8) | 3 (7) | |
| Lower lobe, n (%) | 12 (18) | 6 (14) | |
| Lesion type | | | 0.40 |
| Ring-like, n (%) | 10 (15) | 10 (23) | |
| Web, n (%) | 48 (74) | 27 (63) | |
| Subtotal, n (%) | 5 (8) | 6 (14) | |
| Total occlusion, n (%) | 2 (3) | 0 | |
| Tortuous, n (%) | 0 | 0 | |
| Vessel diameter, mm | 4.49±1.23 | 4.09±1.03 | 0.10 |
| Balloon | | | |
| Size, mm | 4.0 (3.0–5.0) | 4.0 (3.0–4.0) | 0.07 |
| Pressure, atm | 8.0 (6.0–8.0) | 8.0 (8.0–8.0) | 0.01 |
| Balloon-to-Vessel (B/V) ratio | 0.93±0.13 | 0.93±0.15 | 0.92 |

Data are presented as n (%), mean ± SD or median (interquartile ranges). PA, pulmonary artery; POBA, plain old balloon angioplasty; NSE PTA, non-slip element percutaneous transluminal angioplasty.

statistically significant difference (0.526 ± 0.267 in the POBA group, 0.415 ± 0.243 in the NSE PTA group, p = 0.08). There was a significant difference in pressure ratio after BPA (0.766 ± 0.162 in the POBA group, 0.675 ± 0.208 in the NSE PTA group, p = 0.03). Despite this finding, there was no significant difference in the Δ Pressure ratio between the two groups (0.241 ± 0.196 in the POBA group, 0.259 ± 0.177 in the NSE PTA group, p = 0.63) (Fig 4). Because of the negative correlation between the pressure ratio before BPA and the Δ Pressure ratio, we investigated the degree of improvement in the pressure gradient ratio based on the pressure ratio before BPA. The ratio of pressure ratios also had no significant difference between the two groups (1.29 (1.12–2.06) in the POBA group, 1.51 (1.23–2.24) in the NSE PTA group, p = 0.95).

## Predictors of Δ Pressure ratio

We analyzed the predictive factors regarding the improvement of Δ Pressure ratio (Table 4). In univariate analysis, only pressure ratio before BPA demonstrated significant relationships with Δ Pressure ratio. Because the groups were not well matched and pressure gradient before BPA was significantly different at baseline, we explored the usefulness of NSE PTA scoring balloon by multivariate analysis. In multivariate analysis using the multiple linear regression model, only pressure ratio before BPA was associated with Δ Pressure ratio (β coefficient: −0.55, 95% CI: −0.66 to −0.44, P < 0.01). In total, the NSE PTA scoring balloon did not have a significant pressure gradient improvement effect when compared with conventional plain balloon in BPA procedure.

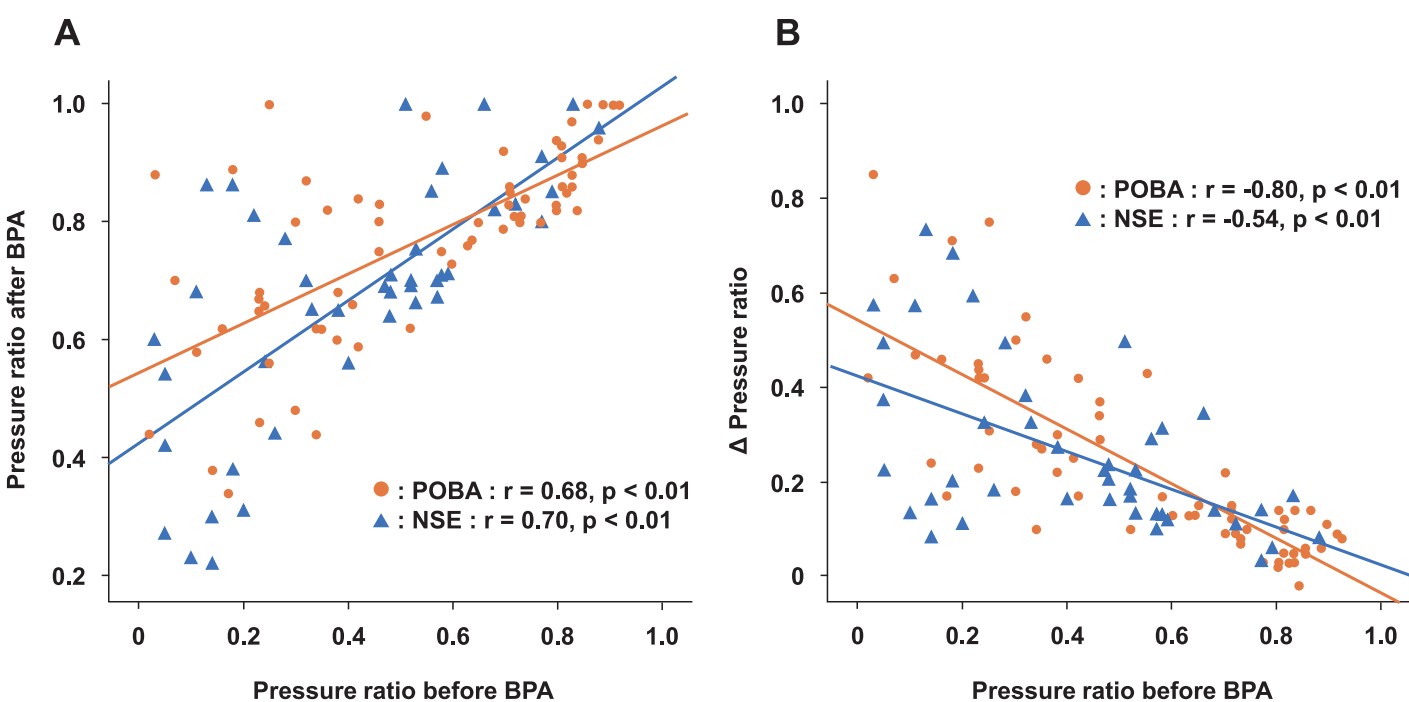

**Fig 3. The relationship between pressure ratio before and after balloon pulmonary angioplasty (BPA).** Pressure ratio before BPA and pressure ratio after BPA plot (A), pressure ratio before BPA and Δ Pressure ratio plot (B).

## Safety of the NSE PTA scoring balloon in BPA procedure

Complications of BPA are presented in Table 5. No complication occurred in the NSE PTA group while 3 hemoptysis episodes were seen in the POBA group due to wire injury, however, there was no significant difference between the two groups (p = 0.27). Additionally, other complications including dissection, perforation, use of noninvasive positive pressure ventilation, need for intubation and death were not seen in either group. At least, the NSE PTA scoring balloon did not increase complications compared to the conventional balloon in BPA.

## Subgroup analysis of Δ Pressure ratio when B/V ratio >1.0

We explored the effectiveness of NSE PTA scoring balloon in the cases where B/V ratio exceeds 1.0 (n = 35) because it has been reported that, in this situation, the effect of the

**Table 3. Pressure ratio of target lesion before and after BPA.**

|  | POBA (n = 65) | NSE PTA (n = 43) | P value |
|---|---|---|---|
| Pressure ratio before BPA | 0.526 ± 0.267 | 0.415 ± 0.243 | 0.08 |
| Pressure ratio after BPA | 0.766 ± 0.162 | 0.675 ± 0.208 | 0.03 |
| Δ Pressure ratio | 0.241 ± 0.196 | 0.259 ± 0.177 | 0.63 |
| Ratio of pressure ratio | 1.29 (1.12–2.06) | 1.51 (1.23–2.24) | 0.95 |

Data are presented as mean ± SD, or median (interquartile ranges). BPA, balloon pulmonary angioplasty; POBA, plain old balloon angioplasty; NSE PTA, non-slip element percutaneous transluminal angioplasty; Δ Pressure ratio, pressure ratio after BPA—pressure ratio before BPA; Ratio of pressure ratio, ratio of pressure ratio after BPA-to-pressure ratio before BPA.

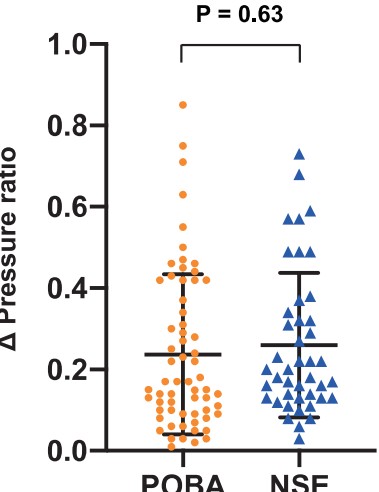

**Fig 4. Δ Pressure ratio in the plain old balloon angioplasty (POBA) group and the non-slip element (NSE) group.** The data are given as the mean ± SD.

scoring balloon may be greater [22]. Univariate analysis showed a significant correlation with the Δ Pressure ratio and the pressure ratio before BPA (beta coefficient: −0.60, 95% CI: −0.80 to −0.41, p < 0.01) (Table 6). However, multivariate analysis showed that the use of the NSE PTA scoring balloon and the pressure ratio before BPA were significantly correlated with Δ Pressure ratio (β coefficient: 0.047, 95% CI: 0.0016 to 0.093, p = 0.043 and β coefficient: -0.60, 95% CI: −0.78 to −0.42, p < 0.01, respectively). Therefore, the NSE PTA scoring balloon shows superior improvement in blood-flow after BPA over the conventional balloon when the B/V ratio is greater than 1.0.

## Discussion

Our findings in this study were as follows: There was no significant difference in Δ Pressure ratio between the POBA group and the NSE PTA group. NSE PTA scoring balloon was safe to use in BPA. In cases where the B/V ratio was higher than 1.0, the Δ Pressure ratio was significantly higher in the NSE PTA group.

CTEPH is known as a progressive disease with a poor prognosis, and with the recent development of BPA, it has been reported that BPA improves pulmonary hypertension, prevents

**Table 4. Univariate and multivariate analysis of Δ Pressure ratio.**

|  | Univariate | | | Multivariate | | |
|---|---|---|---|---|---|---|
|  | β | 95% CI | P value | β | 95% CI | P value |
| Balloon type, NSE | −0.016 | −0.046 to 0.013 | 0.18 | -0.018 | −0.046 to 0.0101 | 0.20 |
| Vessel diameter, mm | −0.077 | −0.17 to 0.016 | 0.063 |  |  |  |
| Balloon Size, mm | 0.082 | −0.023 to 0.19 | 0.10 |  |  |  |
| B/V ratio | −0.19 | −0.65 to 0.28 | 0.37 | 0.16 | −0.0022 to 0.349 | 0.084 |
| Web lesion | 0.022 | −0.0046 to 0.048 | 0.08 |  |  |  |
| Pressure ratio before BPA | **−0.57** | **−0.68 to −0.45** | **<0.01** | **−0.55** | **−0.66 to −0.44** | **<0.01** |

Δ Pressure ratio, pressure ratio after BPA—pressure ratio before BPA; NSE PTA, non-slip element percutaneous transluminal angioplasty; POBA, plain old balloon angioplasty; BPA, balloon pulmonary angioplasty; B/V ration, Balloon-to-Vessel ratio.

**Table 5. Incidence of complications.**

|  | POBA (n = 65) | NSE PTA (n = 43) | P value |
|---|---|---|---|
| Haemoptysis, n (%) | 3 (5) | 0 | 0.27 |
| Dissection, n (%) | 0 | 0 | 1.00 |
| Perforation, n (%) | 0 | 0 | 1.00 |
| NPPV, n (%) | 0 | 0 | 1.00 |
| Intubation, n (%) | 0 | 0 | 1.00 |
| Death, n (%) | 0 | 0 | 1.00 |

Data are presented as n (%). POBA, plain old balloon angioplasty; NSE PTA, non-slip element percutaneous transluminal angioplasty; NPPV, noninvasive positive pressure ventilation.

**Table 6. Univariate and multivariable analysis of Δ Pressure ratio in B/V ratio >1.0 (n = 35).**

|  | Univariate | | | Multivariate | | |
|---|---|---|---|---|---|---|
|  | β | 95% CI | P value | β | 95% CI | P value |
| Balloon type, NSE | 0.057 | −0.0063 to 0.12 | 0.07 | **0.047** | **0.0016 to 0.093** | **0.043** |
| Vessel diameter, mm | 0.23 | −0.36 to 0.83 | 0.43 |  |  |  |
| Balloon Size, mm | −0.21 | −0.77 to 0.34 | 0.44 |  |  |  |
| B/V ratio | 0.523 | −1.15 to 2.20 | 0.53 | −0.088 | −0.5 to 0.28 | 0.57 |
| Web lesion | −0.0001 | −0.047 to 0.047 | 1.0 |  |  |  |
| Pressure ratio before BPA | **−0.60** | **−0.80 to -0.41** | **<0.01** | **−0.60** | **−0.78 to −0.42** | **<0.01** |

Δ Pressure ratio, pressure ratio after BPA—pressure ratio before BPA; NSE PTA, non-slip element percutaneous transluminal angioplasty; POBA, plain old balloon angioplasty; BPA, balloon pulmonary angioplasty; B/V ration, Balloon-to-Vessel ratio.

the development of right heart failure, and improves the prognosis and quality of life of the patient [12, 23, 24]. However, there are still few reports regarding the best strategy for BPA in this population, including the selection of balloons. The NSE PTA scoring balloon is a non-slip peripheral angioplasty catheter with three nylon scoring elements which produce controlled scoring of the vessel wall and work to reduce slipping during balloon inflation. The triangular cross section of the scoring elements provides a higher and more concentrated pressure transfer to the vessel wall, which contributes to the reduction in balloon slipping during expansion. In percutaneous coronary intervention (PCI), the ELEGANT study reported that acute gain was significantly higher than that of normal balloons in in-stent restenosis of coronary arteries [17]. In addition, Soga et al. reported the usefulness of NSE PTA in experimental model was enormously observed in the group with B/V ratio more than 1.0 [22]. In this study, there was no significant difference in Δ Pressure ratio between the NSE PTA group and the POBA group, but the Δ Pressure ratio increased significantly in the NSE PTA group when the B/V ratio was greater than 1.0. This suggests that expansion with an NSE balloon adjusted to the blood vessel diameter is more effective in applying concentrated force due to the non-slip element, which results in an adequate pressure gradient improvement. However, high pulmonary pressure and high perfusion pressure were a risk of reperfusion pulmonary injury [14, 23]. Recently, BPA with undersized balloons and repeated sessions are recommended to get enough improvement with reducing complications [25]. Therefore, NSE PTA may be suitable for consolidation treatment with sufficient low pulmonary artery pressure after several BPA treatments rather than the initial dilation of complete occlusion or severe stenotic lesions with high pulmonary artery pressure. In addition, NSE PTA is more expensive

compared with conventional plain balloon. Therefore, we do not recommend routine use of NSE PTA scoring balloon in BPA procedures, and should be used in the case that we want to dilate pulmonary artery intensively in the last session.

In particular, venous blood vessels are highly compliant, and it is important to select a balloon that is suitable for the diameter of the blood vessel. Initially it was expected that NSE PTA would be more effective for lotus root-like lesions, called web lesions, because the non-slipping element seemed to be forced into the filamentous organized thrombus, but this study did not show a significant difference from conventional balloons whether lesion was a web lesion or not. Because of a small sample size, we could not analyze the treatment effect among the lesion type in detail. Further research will be needed to determine the efficacy of the NSE PTA scoring balloon according to thrombotic lesion type.

Regarding safety concerns, a multicenter registry of BPA in Japan found that complications occurred in 36.3% of patients [9]. In the cases of our study, no procedural complications, such as hemoptysis, vessel dissection or perforation, were observed in the NSE PTA group. Scoring balloon has been reported to reduce the incidence of severe dissection in superficial femoral artery angioplasty [26]. Since the cross section of this element is wedge-shaped, high stress is efficiently concentrated to the vessel wall. This may prevent severe dissection that reaches the media. In addition, because the NSE PTA scoring balloon has three nylon elements, it seems to have less penetrating force into the vessel wall than compared to the cutting balloon with metallic blades. Venous tissue, such as the pulmonary artery, is more fragile than arterial tissue, therefore, NSE PTA scoring balloon might effectively apply pressure and could prevent vascular injury.

There were several limitations in this study. First, this study was conducted retrospectively at a single center with a limited sample size, not a randomized trial. There were several biases including patient and procedure selection. Therefore, both groups are not well matched at baseline. Second, the balloon size, the expansion pressure and the choice of NSE PTA scoring balloon or POBA were at the discretion of the operator. A multicenter and randomized trial with a larger number of patients is needed to validate our findings. Third, since BPA is often performed on many lesions during one session, it is not possible to compare major adverse cardiovascular events and the improvement of mean pulmonary artery pressure by balloon selection alone. In addition, we evaluated only immediately after BPA. It has been reported that treated pulmonary arteries may take time to dilate sufficiently. We need to investigate follow-up data to conclude the effectiveness of scoring balloon in BPA procedures. Finally, the pressure gradient depends on peripheral vascular resistance caused by vasculopathy. We could not put the presence of vasculopathy in the analysis as a confounder.

## Conclusion

NSE PTA scoring balloon did not show a significant improvement in pressure gradient compared to conventional balloon. Nevertheless, NSE PTA scoring balloon showed effective blood-flow improvement in the case of large balloon-to-vessel ratio. NSE PTA scoring balloon was safe to use in BPA procedure. Our results suggest that the use of the NSE PTA scoring balloon should be considered as one of the treatment options for BPA, especially in the case that we want to dilate pulmonary artery intensively in the last session.

## Acknowledgments

We would like to thank Dr. Sean Delue for proofreading in English. We gratefully acknowledge the work of past and present members of our hospital.

## Author Contributions

**Conceptualization:** Hideo Tsubata, Naohiko Nakanishi.

**Data curation:** Masao Takigami, Hideo Tsubata, Naohiko Nakanishi, Yuki Matsubara, Noriyuki Wakana, Kenji Yanishi.

**Formal analysis:** Masao Takigami, Naohiko Nakanishi.

**Investigation:** Hideo Tsubata, Naohiko Nakanishi, Yuki Matsubara, Noriyuki Wakana, Kenji Yanishi.

**Methodology:** Hideo Tsubata, Naohiko Nakanishi.

**Project administration:** Naohiko Nakanishi.

**Supervision:** Naohiko Nakanishi, Kan Zen, Takeshi Nakamura, Satoaki Matoba.

**Validation:** Naohiko Nakanishi.

**Visualization:** Masao Takigami, Naohiko Nakanishi.

**Writing – original draft:** Masao Takigami.

**Writing – review & editing:** Naohiko Nakanishi.

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
