## [Decision Letter · Decision Letter 0]

10 Dec 2021

PONE-D-21-25699The effectiveness of scoring balloon angioplasty in the treatment of chronic thromboembolic pulmonary hypertensionPLOS ONE

Dear Dr. Nakanishi,

Thank you for submitting your manuscript to PLOS ONE. After careful consideration, we feel that it has merit but does not fully meet PLOS ONE’s publication criteria as it currently stands. Therefore, we invite you to submit a revised version of the manuscript that addresses the points raised during the review process.

Both of the reviewers regard this experiment important, so please revise the manuscript.

We look forward to receiving your revised manuscript.

Kind regards,

Yoshiaki Taniyama, MD, PhD

Academic Editor

PLOS ONE

Journal Requirements:

2. In your ethics statement in the manuscript and in the online submission form, please ensure that you have discussed whether all data/samples were fully anonymized before you accessed them and/or whether the IRB or ethics committee waived the requirement for informed consent. If patients provided informed written consent to have data/samples from their medical records used in research, please include this information.

Reviewers' comments:

Reviewer's Responses to Questions

**Comments to the Author**

1. Is the manuscript technically sound, and do the data support the conclusions?

Reviewer #1: Yes

Reviewer #2: No

2. Has the statistical analysis been performed appropriately and rigorously? 

Reviewer #1: Yes

Reviewer #2: No

3. Have the authors made all data underlying the findings in their manuscript fully available?

Reviewer #1: Yes

Reviewer #2: No

4. Is the manuscript presented in an intelligible fashion and written in standard English?

Reviewer #1: Yes

Reviewer #2: Yes

5. Review Comments to the Author

Reviewer #1: Plain old balloon angioplasty (POBA) group vs the non-slip element (NSE) manuscript review.

Thank you so very much for the opportunity to review this interesting manuscript.

I am in agreement that we need to explore options for best delivery of balloon angioplasty to our chronic thromboembolic pulmonary hypertension patients.

I found this very interesting and helpful.

I do agree that additional study on a larger scale would be very helpful.

Your description of methods and statistics for sound and I do not recommend any significant change.

Comments regarding cost and implications regarding that may be helpful to the reader. That may be something to consider to add regarding catheter choice.

I agree with your comments and findings regarding effective blood flow improvement in the case of large balloon two-vessel ratio is something that needs to be explored further in the future as catheter selection may be based on pre-BPA assessment and does require further study to determine if this would improve outcomes.

I do recommend to review your conclusions as I think expanding this could be helpful to the reader.

Conclusion

2 NSE PTA scoring balloon did not show a significant improvement in pressure

3 gradient compared to conventional balloon. Nevertheless, NSE PTA scoring

4 balloon showed effective blood-flow improvement in the case of large balloon-to-

5 vessel ratio. NSE PTA scoring balloon was safe to use in BPA procedure. Our

6 results suggest that the use of the NSE PTA scoring balloon should be considered

7 as one of the treatment options for BPA.

Additional comments....................

When reading I did find a few typos noted below:

Page 10

6 pressure microcatheter on the guidewire was advanced to the tip of guiding

7 catheter where the 2 pressures, a tip of guiding catheter and pressure

8 microcatheter, were equalized after saline flash.

**FLUSH

Page 10

Δ Pressure ratio was defined as the difference between the pressure

13 ratio before and after BPA. Finally, the ratio of pressure ratios was defined as the

14 ratio between the before BPA pressure ratio and the after BPA pressure ration.

**RATIO

Question:

Page 13

11 was better controlled; mean right atrium pressure (RAP) was 8.0 ± 4.0 mmHg,

12 and pulmonary artery wedge pressure (PAWP) was 9.4 ± 3.5 mmHg.

**THIS NOTES BETTER CONTROLLED but likely should just be controlled as not comparing

Page 21

11 95% CI: -0.76 to -0.43, p < 0.01, respectively). Therefore, the NSE PTA scoring

12 balloon shows superior improvement in blood-flow after BPA over the

13 conventional balloon when the B/V ration is greater than 1.0.

**RATIO

Page 24

1 In addition, Soga et al. reported the usefulness of NSE PTA in experimental model

2 was enormously observed in the group with B/V ratio more than 1.0 [22].

**Would replace ENORMOUSLY here.

Page 25

Since the cross section of this element is wedge-shaped, high stress is

13 efficiently concentrated to the vessel wall. This may prevent severe dissociation

14 that reaches the media.

**DISSECTION

Page 26

1 There were several limitations in this study. First, this study was

2 conducted retrospectively at a single center with a limited sample size, not

3 randomized trial.

**A RANDOMONIZED TRIAL

Page 26

. In addition, we evaluated

11 only immediately after BPA. It has been reported that treated pulmonary arteries

12 are took time to dilate sufficiently.

**MAY TAKE TIME

Page 26

We need to investigate follow-up data to

13 conclude the effectiveness of scoring balloon in BPA procedure.

**PROCEDURES

Reviewer #2: PONE-D-21-25699: statistical review

SUMMARY. This is a retrospective study that compares the effect of the non-slip element percutaneous transluminal angioplasty (NSE PTA) on blood pressure ratios to the effect of plain-balloon treatments (POBA). The statistical analysis relies on t-test methods for comparing the NSE PTA and the POBA groups and on regression analysis for evaluating the effect of the use of NSE PTA scoring balloon on the pressure ratio in a subset of data where the balloon-to-vessel ratio exceeded 1.0. I have several major concerns about the statistical methods that have been employed in this paper.

MAJOR ISSUES

1. The statistical analysis focuses on the improvements of pressure ratios observed after Balloon pulmonary angioplasty (BPA) in 108 lesions, which are treated as independent observations. However, if I understood correctly, these lesions are clustered within 14 subjects and, as such, they may not be assumed as independent cases. Instead, they should be treated as repeated measures within a sample of 14 subjects. Repeated measures methods are available in SAS, which is the software used by the authors. Notice that the repeated measures structure of the data must be accounted for in both the ANOVA analysis that compares the NSE PTA and the POBA groups and in the regression analysis that focuses on the effect of the NSE PTA treatment.

2. This is a retrospective study and, as a result, subjects have not been randomly associated to the NSE PTA and the POBA groups. The authors should therefore provide evidence that these two groups do not significantly differ with respect to the biometrical variables that are summarized in Table 1. Otherwise, results could be biased by group-specific differences

3. The statistical analysis relies on the assumption that the dependent variables are normally distributed. The authors should provide some evidence that such assumption is realistic, otherwise all the p-values of the paper could be biased.

6. PLOS authors have the option to publish the peer review history of their article (what does this mean?). If published, this will include your full peer review and any attached files.

Reviewer #1: No

Reviewer #2: No

---

## [Author Response · Author response to Decision Letter 0]

26 Dec 2021

Response to Reviewer comments (PONE-D-21-25699):  

  

We would like to thank the Editors and Reviewers for their comments on our manuscript, including their very useful suggestions for improvement. We have endeavored to respond to their comment and suggestions, and we believe that we have adequately modified the revised manuscript to address their concerns.

Responses to all queries are highlighted with Track Changes in the revised manuscript.

Reviewer #1:

Plain old balloon angioplasty (POBA) group vs the non-slip element (NSE) manuscript review.

Thank you so very much for the opportunity to review this interesting manuscript.

I am in agreement that we need to explore options for best delivery of balloon angioplasty to our chronic thromboembolic pulmonary hypertension patients. I found this very interesting and helpful. I do agree that additional study on a larger scale would be very helpful. Your description of methods and statistics for sound and I do not recommend any significant change. Comments regarding cost and implications regarding that may be helpful to the reader. That may be something to consider to add regarding catheter choice. I agree with your comments and findings regarding effective blood flow improvement in the case of large balloon two-vessel ratio is something that needs to be explored further in the future as catheter selection may be based on pre-BPA assessment and does require further study to determine if this would improve outcomes. 

I do recommend to review your conclusions as I think expanding this could be helpful to the reader.

Conclusion

NSE PTA scoring balloon did not show a significant improvement in pressure gradient compared to conventional balloon. Nevertheless, NSE PTA scoring balloon showed effective blood-flow improvement in the case of large balloon-to-vessel ratio. NSE PTA scoring balloon was safe to use in BPA procedure. Our results suggest that the use of the NSE PTA scoring balloon should be considered as one of the treatment options for BPA.

Response:

We very much appreciate your valuable comments and suggestions. As you mentioned, the cost-benefit is an important issue of balloon choice. NSE PTA is more expensive compared with conventional plain balloon. Therefore, we do not recommend routine use of NSE PTA scoring balloon in BPA procedures, and should be used in the case that we want to dilate pulmonary artery intensively in the last session. According to your comments, we have revised the manuscript as follows.

(Discussion, p24, line 15-18)

In addition, NSE PTA is more expensive compared with conventional plain balloon. Therefore, we do not recommend routine use of NSE PTA scoring balloon in BPA procedures, and should be used in the case that we want to dilate pulmonary artery intensively in the last session.

(Conclusion, p28, line 5-8)

Our results suggest that the use of the NSE PTA scoring balloon should be considered as one of the treatment options for BPA, especially in the case that we want to dilate pulmonary artery intensively in the last session.

Additional comments....................

When reading I did find a few typos noted below:

Page 10

pressure microcatheter on the guidewire was advanced to the tip of guiding　catheter where the 2 pressures, a tip of guiding catheter and pressure　microcatheter, were equalized after saline flash.

**FLUSH

Response:

Thanks to your pointing. We corrected it as below:

(Methods, p10, line 5-8)

The pressure microcatheter on the guidewire was advanced to the tip of guiding catheter where the 2 pressures, a tip of guiding catheter and pressure microcatheter, were equalized after saline flush.

Page 10

ΔPressure ratio was defined as the difference between the pressure　ratio before and after BPA. Finally, the ratio of pressure ratios was defined as the　ratio between the before BPA pressure ratio and the after BPA pressure ration.

**RATIO

Response:

We greatly appreciate your comment. We corrected it as below:

(Methods, p10, line 12-14)

Δ Pressure ratio was defined as the difference between the pressure ratio before and after BPA. Finally, the ratio of pressure ratios was defined as the ratio between the before BPA pressure ratio and the after BPA pressure ratio.

Question:

Page 13

was better controlled; mean right atrium pressure (RAP) was 8.0 ± 4.0 mmHg, and pulmonary artery wedge pressure (PAWP) was 9.4 ± 3.5 mmHg.

**THIS NOTES BETTER CONTROLLED but likely should just be controlled as not comparing

Response:

We thank to your pointing. As you mentioned, these are not compared data and we corrected it as below:

(Results, p13, line 10-12)

Fluid balance was well controlled; mean right atrium pressure (RAP) was 8.0 ± 4.0 mmHg, and pulmonary artery wedge pressure (PAWP) was 9.4 ± 3.5 mmHg.

Page 21

95% CI: -0.76 to -0.43, p < 0.01, respectively). Therefore, the NSE PTA scoring balloon shows superior improvement in blood-flow after BPA over the conventional balloon when the B/V ration is greater than 1.0.

**RATIO

Response:

Thanks to your pointing. We corrected it as below:

(Results, p21, line 8-10)

Therefore, the NSE PTA scoring balloon shows superior improvement in blood-flow after BPA over the conventional balloon when the B/V ratio is greater than 1.0.

Page 24

In addition, Soga et al. reported the usefulness of NSE PTA in experimental model was enormously observed in the group with B/V ratio more than 1.0 [22].

**Would replace ENORMOUSLY here.

Response:

We greatly appreciate your comment. We deleted ‘ENORMOUSLY’ as below:

(Discussion, p24, line 1-2)

In addition, Soga et al. reported the usefulness of NSE PTA in experimental model was observed in the group with B/V ratio more than 1.0 [22].

Page 25

Since the cross section of this element is wedge-shaped, high stress is efficiently concentrated to the vessel wall. This may prevent severe dissociation that reaches the media.

**DISSECTION

Response:

We thank to your pointing. We modified it as below:

(Discussion, p25, line 15-17)

Since the cross section of this element is wedge-shaped, high stress is efficiently concentrated to the vessel wall. This may prevent severe dissection that reaches the media.

Page 26

There were several limitations in this study. First, this study was conducted retrospectively at a single center with a limited sample size, not randomized trial.

**A RANDOMONIZED TRIAL

Response:

We greatly appreciate your comment. We corrected it as below:

(Discussion, p26, line 4-6)

There were several limitations in this study. First, this study was conducted retrospectively at a single center with a limited sample size, not a randomized trial.

Page 26

. In addition, we evaluated only immediately after BPA. It has been reported that treated pulmonary arteries are took time to dilate sufficiently.

**MAY TAKE TIME

Page 26

We need to investigate follow-up data to conclude the effectiveness of scoring balloon in BPA procedure.

**PROCEDURES

Response:

Thanks to your pointing. We modified it as below:

(Discussion, p26, line 14-16)

It has been reported that treated pulmonary arteries may take time to dilate sufficiently. We need to investigate follow-up data to conclude the effectiveness of scoring balloon in BPA procedures.

Reviewer #2: PONE-D-21-25699: statistical review

SUMMARY. This is a retrospective study that compares the effect of the non-slip element percutaneous transluminal angioplasty (NSE PTA) on blood pressure ratios to the effect of plain-balloon treatments (POBA). The statistical analysis relies on t-test methods for comparing the NSE PTA and the POBA groups and on regression analysis for evaluating the effect of the use of NSE PTA scoring balloon on the pressure ratio in a subset of data where the balloon-to-vessel ratio exceeded 1.0. I have several major concerns about the statistical methods that have been employed in this paper.

Response:

We thank Reviewer #2 for the kind and constructive comments. I have described below our responses to the reviewers' comments and the changes to the revised manuscript, tables and figures. I hope that these responses will satisfy Reviewer #2.

MAJOR ISSUES

1. The statistical analysis focuses on the improvements of pressure ratios observed after Balloon pulmonary angioplasty (BPA) in 108 lesions, which are treated as independent observations. However, if I understood correctly, these lesions are clustered within 14 subjects and, as such, they may not be assumed as independent cases. Instead, they should be treated as repeated measures within a sample of 14 subjects. Repeated measures methods are available in SAS, which is the software used by the authors. Notice that the repeated measures structure of the data must be accounted for in both the ANOVA analysis that compares the NSE PTA and the POBA groups and in the regression analysis that focuses on the effect of the NSE PTA treatment.

Response:　

I appreciate your constructive comments. In the BPA procedures, many pulmonary branches are treated in one patient in several sessions. In this study, 108 lesions are different branches in 14 patients. Since each lesion were dilatated by either plain balloon or NSE PTA balloon, we consider these lesions were independent respectively. However, as you mentioned, these lesions were treated repeatedly in same patients. Therefore, we used mixed models for repeated measures as you pointed out, and confirmed the almost same results. We corrected the revised tables and manuscript.

Manuscript:

(Method, p11, line 8-11): 

We compared the baseline characteristics, pressure ratio and Δ Pressure ratio at the target lesion between the NSE PTA group and the POBA group before and after BPA procedure using mixed models for repeated measures.

(Method, p11, line 15-18): 

We performed univariate analysis using regression analysis with mixed models for repeated measures to find the relationship of Δ Pressure ratio in whole patients and patients with B/V ratio over 1.0 according to Soga's report [22].

(Result, p16-17, line 16-4): 

Although pressure ratio before BPA tended to be lower in the NSE PTA group, there was not statistically significant difference (0.526 ± 0.267 in the POBA group, 0.415 ± 0.243 in the NSE PTA group, p = 0.08). There was a significant difference in pressure ratio after BPA (0.766 ± 0.162 in the POBA group, 0.675 ± 0.208 in the NSE PTA group, p = 0.03). Despite this finding, there was no significant difference in the Δ Pressure ratio between the two groups (0.241 ± 0.196 in the POBA group, 0.259 ± 0.177 in the NSE PTA group, p = 0.63)

(Result, p18, line 11-12): 

In univariate analysis, only pressure ratio before BPA demonstrated significant relationships with Δ Pressure ratio.

2. This is a retrospective study and, as a result, subjects have not been randomly associated to the NSE PTA and the POBA groups. The authors should therefore provide evidence that these two groups do not significantly differ with respect to the biometrical variables that are summarized in Table 1. Otherwise, results could be biased by group-specific differences

Response:

We apologize for confusing you. In the BPA procedures, many pulmonary branches are treated in one patient in several sessions. Each vessel was treated with either POBA or NSE PTA. Therefore, both plain balloon and NSE PTA balloon were used in same patient. We added additional information about BPA procedures and lesion selection that were analyzed as follows.

(Abstract, p3, line 7-9)

108 pulmonary artery branches in 14 CTEPH patients who underwent BPA using NSE PTA scoring balloon (the NSE PTA group) or plain balloon (the POBA group) and pressure gradient evaluation were analyzed.

(Method, p8, line 5-7)

108 pulmonary artery branches in 14 consecutive patients with CTEPH who underwent BPA using NSE PTA scoring balloon or plain balloon and had pressure gradient evaluation with pressure microcatheter were analyzed.

3. The statistical analysis relies on the assumption that the dependent variables are normally distributed. The authors should provide some evidence that such assumption is realistic, otherwise all the p-values of the paper could be biased.

Response:

We greatly appreciate your comment. As a result of the verification, we consider that all variables except ‘Balloon size’, ‘Balloon Pressure’, and ‘Ratio of pressure ratio’ are normally distributed from each histogram and Q-Q plot. As a representative validation, the histogram and the Q-Q plot of ΔPressure ratio are shown below. Valuables that were not normally distributed were analyzed by using nonparametric analysis. According to your recommendation, we revised our manuscript about the evaluation of normal distribution and Tables.

(Method, p11, line 6)

We evaluated whether each valuable was normally distributed.

(Method, p11, line 12-13)

Valuables that were not normally distributed were compared using non-parametric test.

(Result, p14, line 1-4)

Although vessel diameter and balloon size tended to be smaller in the NSE PTA group, there did not appear to be statistically significant difference. Balloon pressure was higher in the NSE PTA group (8.0 (6.0–8.0) atm in the POBA group and 8.0 (8.0–8.0) atm in the NSE PTA group, p = 0.01).

---

## [Decision Letter · Decision Letter 1]

17 Jan 2022

The effectiveness of scoring balloon angioplasty in the treatment of chronic thromboembolic pulmonary hypertension

PONE-D-21-25699R1

Dear Dr. Nakanishi,

We’re pleased to inform you that your manuscript has been judged scientifically suitable for publication and will be formally accepted for publication once it meets all outstanding technical requirements.

Kind regards,

Yoshiaki Taniyama, MD, PhD

Academic Editor

PLOS ONE

Additional Editor Comments (optional):

The author responded the problems.

Reviewers' comments:

Reviewer's Responses to Questions

**Comments to the Author**

1. If the authors have adequately addressed your comments raised in a previous round of review and you feel that this manuscript is now acceptable for publication, you may indicate that here to bypass the “Comments to the Author” section, enter your conflict of interest statement in the “Confidential to Editor” section, and submit your "Accept" recommendation.

Reviewer #2: All comments have been addressed

2. Is the manuscript technically sound, and do the data support the conclusions?

Reviewer #2: (No Response)

3. Has the statistical analysis been performed appropriately and rigorously? 

Reviewer #2: (No Response)

4. Have the authors made all data underlying the findings in their manuscript fully available?

Reviewer #2: (No Response)

5. Is the manuscript presented in an intelligible fashion and written in standard English?

Reviewer #2: (No Response)

6. Review Comments to the Author

Reviewer #2: (No Response)

7. PLOS authors have the option to publish the peer review history of their article (what does this mean?). If published, this will include your full peer review and any attached files.

Reviewer #2: No

---

## [Editor Report · Acceptance letter]

20 Jan 2022

PONE-D-21-25699R1 

The effectiveness of scoring balloon angioplasty in the treatment of chronic thromboembolic pulmonary hypertension 

Dear Dr. Nakanishi:

I'm pleased to inform you that your manuscript has been deemed suitable for publication in PLOS ONE. Congratulations! Your manuscript is now with our production department. 

Kind regards, 

on behalf of

Dr. Yoshiaki Taniyama 

Academic Editor

PLOS ONE